# Cell specific delivery of modified mRNA expressing therapeutic proteins to leukocytes

Nuphar Veiga[1,2,3,4], Meir Goldsmith[1,2,3,4], Yasmin Granot[1,2,3,4], Daniel Rosenblum[1,2,3,4], Niels Dammes[1,2,3,4], Ranit Kedmi[1,2,3,4,5], Srinivas Ramishetti[1,2,3,4] & Dan Peer [1,2,3,4]

Therapeutic alteration of gene expression in vivo can be achieved by delivering nucleic acids (e.g., mRNA, siRNA) using nanoparticles. Recent progress in modified messenger RNA (mmRNA) synthesis facilitated the development of lipid nanoparticles (LNPs) loaded with mmRNA as a promising tool for in vivo protein expression. Although progress have been made with mmRNA-LNPs based protein expression in hepatocytes, cell specificity is still a major challenge. Moreover, selective protein expression is essential for an improved therapeutic effect, due to the heterogeneous nature of diseases. Here, we present a precision protein expression strategy in Ly6c$^+$ inflammatory leukocytes in inflammatory bowel disease (IBD) induced mice. We demonstrate a therapeutic effect in an IBD model by targeted expression of the interleukin 10 in Ly6c$^+$ inflammatory leukocytes. A selective mmRNA expression strategy has tremendous therapeutic potential in IBD and can ultimately become a novel therapeutic modality in many other diseases.

[1] Laboratory of Precision NanoMedicine, School of Molecular Cell Biology and Biotechnology, George S. Wise Faculty of Life Sciences, Tel Aviv 69978, Israel. [2] Department of Materials Sciences and Engineering, Iby and Aladar Fleischman Faculty of Engineering, Tel Aviv 69978, Israel. [3] Center for Nanoscience and Nanotechnology, Tel Aviv 69978, Israel. [4] Cancer Biology Research Center, Tel Aviv University, Tel Aviv 69978, Israel. [5] Present address: Molecular Pathogenesis Program, The Kimmel Center for Biology and Medicine of the Skirball Institute, New York University School of Medicine, New York, NY 10016, USA. These authors contributed equally: Nuphar Veiga, Meir Goldsmith. Correspondence and requests for materials should be addressed to D.P. (email: peer@tauex.tau.ac.il)

Extensive research in the last few years has emphasized dysregulation of gene expression in various pathologies such as cancer, inflammatory disorders, deficiency syndromes and neurodegenerative diseases. Therefore, the ability to specifically manipulate gene expression either by overexpression of a desired protein using stable, modified messenger RNAs (mmRNA), or by short interfering RNAs (siRNAs) that mediate gene silencing in the desired tissues or cells, holds great promise for therapeutic applications. However, the use of mRNA molecules for expressing a desired protein has been hindered due to technological challenges, such as the ability to successfully transcribed in vitro mRNA in large amounts, instability in vivo, and immunogenicity. Recent insights into mRNA structure and function, together with the advances in in vitro transcription methods and the introduction of modified nucleotides (e.g., 5mC, pseudo-Uridine etc.), facilitate the utilization of mRNA for therapeutic applications with higher expression efficiencies and lower immunogenicity[1–3]. Nevertheless, both strategies to manipulate protein expression require the untrivial intracellular delivery of RNA molecules. Furthermore, a critical challenge of RNA-based therapeutic approach lies in the ability to deliver RNA molecules effectively to specific target cells.

These challenges promoted the development of synthetic and natural delivery systems as a promising strategy for non-viral and viral gene manipulation, respectively. Although viral vectors demonstrate efficient nucleic acids delivery, immunogenicity and safety concerns might hinder their utilization for long term therapeutics. In contrast, lipid nanoparticles (LNPs) were designed as efficient, non-immunogenic and a safe alternative system for in vivo gene manipulation. LNPs protect RNA molecules from degradation and immune activation and facilitate their cellular uptake and release from endosomal compartments to the cytosol. Furthermore, utilizing LNPs for RNA delivery has become more feasible as encapsulation efficiency was improved with the utilization of pH-dependent ionizable lipids and the implementation of relevant microfluidic production methodologies[4–8]. Those developments lead to a robust and uniform production of LNPs, reduced immunogenicity and improved the release of RNA molecules in the cytoplasm[9].

Although mRNA loaded LNPs have been utilized for vaccinations in local administration[2] and for hepatocyte based protein expression in systemic administration[10,11], systemic, cell-specific targeting of mRNA molecules remains a challenge[12]. To overcome the hurdles of selective, targeted delivery of lipid-based technologies, we recently developed a modular targeting platform named ASSET (Anchored Secondary scFv Enabling Targeting). ASSET coats the LNPs with monoclonal antibodies (mAbs) and enables a flexible switching between different targeting mAbs. ASSET utilizes a biological approach and facilitates the construction of a theoretically unlimited repertoire of targeted carriers, which deliver RNA molecules efficiently to various leukocytes subsets in vivo[7].

Inflammatory bowel disease (IBD) is characterized by a complex and dysregulated immune response. The onset of IBD is considered as a combination of genetic alterations and environmental factors. Consequently, an effective treatment for IBD requires a temporal and spatial immunosuppressive effect that will tune down the autoimmune response without effecting normal immune activity. Immune response can be adjusted by altering the local concentrations of immune modulating signal molecules such as cytokines. Cytokines are potent immunomodulation proteins with pro and/or anti-inflammatory effects on the immune system. Among them, Interleukin 10 (IL10) is one of the most central anti-inflammatory cytokine involved in balancing intestinal immunity. Despite the potential, clinical trials using recombinant IL10 (rIL10) failed to demonstrate a robust therapeutic effect. The main hurdles with rIL10 treatment were the short half-life of only a few hours and the high protein concentration needed for systemic administration, which resulted in increased toxicity[13–15]. As an alternative, gene therapy enables tailored gene manipulation, as well as lower production limitations and efficient protein production with proper post transcription modification. For instance, adeno-viral mediated IL10 protein expression in vivo was proven previously as a valuable approach for controlling the immune response in IBD[16–18]. Although impressive therapeutic effect was demonstrated using Adeno-virus based IL10 expression, it cannot be given more than once due to safety limitations and a safer alternative is required for clinical translation. Furthermore, another advantage of mRNA-based therapies is the lower risk for genomic integration, which is a major risk factor in DNA based treatments.

Herein, we demonstrated an efficient mRNA encapsulation into LNPs, followed by a significant cell specific protein expression in vitro and in vivo. We utilized mRNA loaded LNPs, combined with the ASSET platform, for targeted gene expression in Ly6c[+] inflammatory leukocytes and achieved a selective protein expression in vivo with a favorable bio-distribution. Finally, we evaluated the therapeutic potential of cell specific expression of the anti-inflammatory IL10 cytokine in Ly6c[+] inflammatory leukocytes in a dextran sodium sulfate (DSS) colitis model. This study opens new avenues in cell specific delivery of mRNA molecules and ultimately might introduce IL-10 mRNA as a novel therapeutic modality to inflammatory bowel diseases.

## Results

**LNPs' optimization for mmRNA encapsulation and delivery**. The formulation of LNPs was designed and optimized to enable a controllable mmRNA encapsulation and to promote an efficient gene expression in leukocytes. mmRNA was formulated in LNPs using the NanoAssemblr® microfluidic mixing system (Precision Nanosystems Inc., Vancouver, Canada), in which mRNA molecules self-assembled with ionizable lipids in acidic condition to form highly uniform nanoparticles, of $63.7 \pm 1.59$ nm in diameter (average size ± s.d., Fig. 1a, schematic illustration, Supplementary Table 1, $n = 5$). To further characterize the ultrastructure of mmRNA LNPs, transmission electron microscopy was performed (Fig. 1b) and encapsulation efficiency was determined by fluorescent RNA detection dye to be ~100% (Fig. 1c, Supplementary Figure 1a-b). We have evaluated the ability of mmRNA LNPs to promote protein expression in vitro in Raw 264.7 macrophages cell line (Supplementary Figure 1c−d) and ex-vivo in mouse splenocytes. We have demonstrated sustained protein expression ex-vivo in mouse splenocytes of both Firefly luciferase (fLuc) (Fig. 1d) and IL10 upon treatment with mmRNA LNPs for up to 48 h (Fig. 1e).

**Endowing cell specificity to mmRNA-LNPs by ASSET platform**. To enable cell specific expression of mmRNA delivered by LNPs in Ly6c[+] inflammatory leukocytes, we have coated LNPs entrapping mmRNA with either anti-Ly6c or IgG control mAbs using our recently developed ASSET linker strategy[7]. For a precise protein expression in specific cell type in vivo, we choose the DSS induced colitis model where Ly6c[+] leukocytes play a major role in the inflammatory disease[19]. ASSET molecules were self-assembled into the LNPs lipid layer and primary antibodies were integrated as previously described[7], forming LNPs coated with anti-Ly6c mAbs for cell-specific delivery of mmRNAs (Fig. 1f, schematic illustration). ASSET incorporation and the addition of targeting Abs did not hamper mRNA encapsulation, LNPs size distribution or ultrastructure (Fig. 1g, h and Supplementary Table 1).

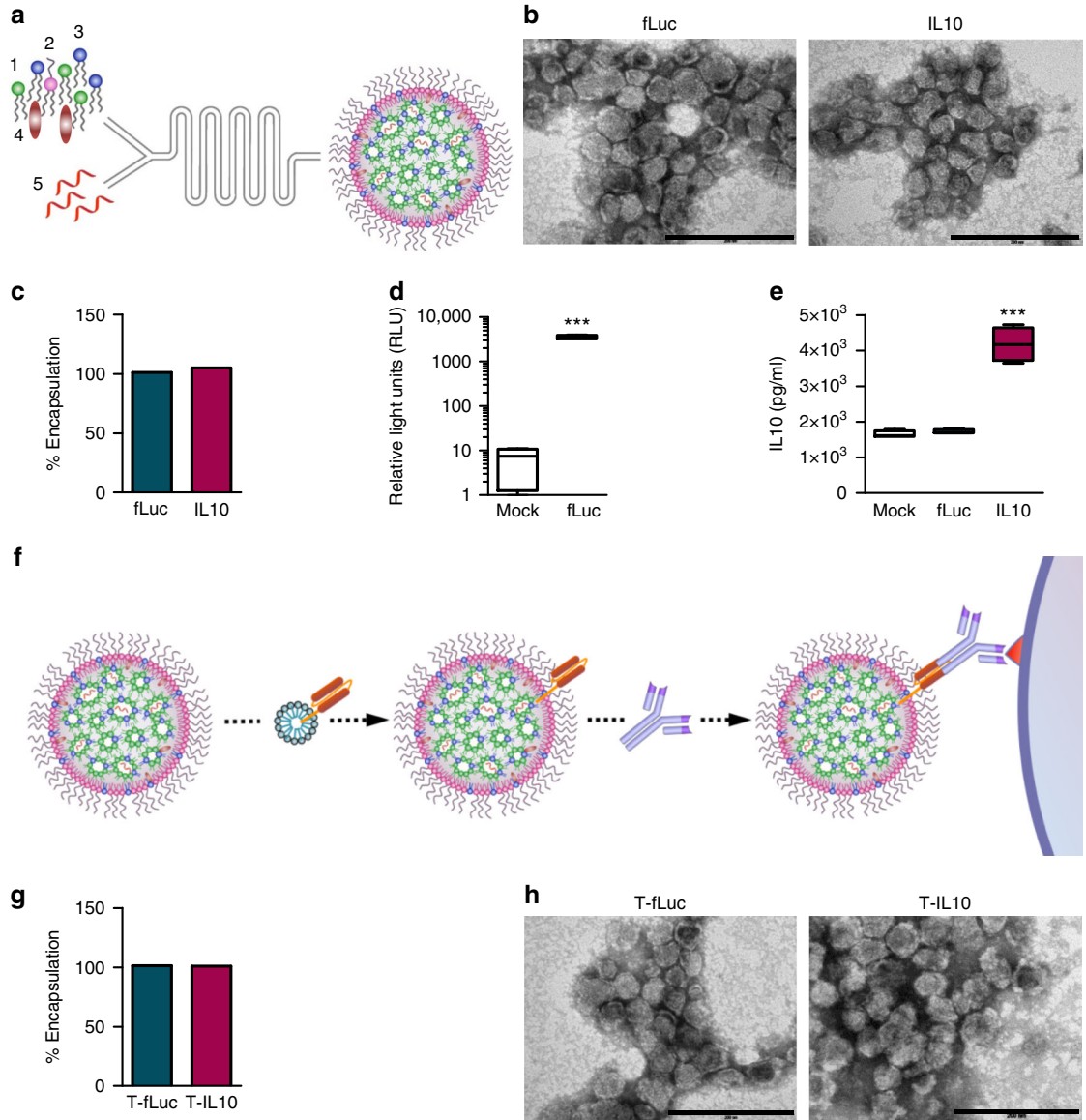

**Fig. 1** mmRNA–tLNPs construction and characterization. **a** Schematic illustration of LNPs preparation. A microfluidic based mixing of lipids, helper lipid DSPC (1), PEG-lipid (2), ionizable amino lipid DLin-MC3-DMA (3) and cholesterol (4), diluted in ethanol and mmRNA(5) diluted in acidic acetate buffer (pH 4) to construct LNPs encapsulated with mmRNA. Upon mixing in acidic condition, electrostatic interactions drive the formation of inverted micelles containing mmRNA molecules surrounded predominantly by ionizable lipid, which are then coated with PEG-lipids to form the LNPs. After LNPs formation, the pH is raised to physiological pH, leading to LNPs with a biocompatible neutral charge. Subsequently, upon endocytosis to the acidic endosome, the amino group of the DLin-MC3-DMA becomes positively charged, which allows the release of mmRNA molecules to the cytosol. **b** Representative transmission electron microscopy (TEM) images of LNP encapsulating fLuc mmRNA (left) and IL10 mmRNA (right). **c** Encapsulation efficiency as measured using RiboGreen assay. Free mmRNA concentration in Triton-permeabilized LNPs solution divided by free mmRNA concentration in intact LNPs solution. **d** Ex-vivo fLuc expression in leukocytes isolated from mouse spleen, 24 h after incubation with fLuc mmRNA LNPs (5 μg/ml mmRNA). Data is presented as Relative Light Units (RLU) per $10^5$ cells. **e** Ex-vivo IL10 expression in leukocytes isolated from mouse spleen, 48 h after incubation with fLuc and IL10 mmRNA LNPs (5 μg/ml mmRNA). Presented as pg/ml, secreted from a $2 \times 10^6$ cells/ml culture. Two-sided Student's $t$-test comparing fLuc to IL10. **f** Schematic illustration of the introduction of targeting moiety to mmRNA loaded LNPs. LNPs are mixed with ASSET micelles and coated with Rat IgG2a primary mAbs, which promote a selective binding of tLNPs to the target receptor. **g** Encapsulation efficiency as measured using RiboGreen assay. Free mmRNA concentration in Triton-permeabilized tLNPs solution divided by free mmRNA concentration in intact tLNPs solution. **h** Representative TEM images of tLNP encapsulating fLuc mmRNA (left) and IL10 mmRNA (right). Data are mean ± s.d. **c–g** or Interquartile range (IQR) with a median center line and min to max error bars (**d**, **e**), $n = 5$, ***$p < 0.001$, two-sided Student's $t$-test **c–e**, **g** Data are representative of three independent experiments

**Specific protein expression in Ly6c$^+$ leukocytes in vivo.** The next set of experiments aimed to evaluate the specificity of the targeted lipid nanoparticles to promote protein expression in Ly6c$^+$ leukocytes in vivo. We employed the fLuc reporter mmRNA-targeted tLNPs to evaluate the bio-distribution in vivo, by a reinforced enzymatic reaction with luciferin, using the IVIS imaging system. Anti-Ly6c tLNPs (T-fLuc) or IgG control LNPs (I-fLuc) fLuc mmRNA-LNPs were injected intravenously to DSS induced mice. fLuc bio-luminescence in the liver, spleen, and intestine was analyzed 24 h post-administration. Analysis revealed ~20-fold increase of fLuc expression in the intestine, a significant increase in the spleen and ~10-fold lower fLuc

expression in the liver using T-fLuc compared to I-fLuc (Fig. 2a–f, Supplementary Table 2). Intestinal fLuc reporter gene was expressed in a segmental manner, which we believe correlates with the segmented nature of the intestinal inflammation in DSS colitis model. We did not observe notable differences between distal and proximal regions of the intestine and the colon (Fig. 2e, f). To further evaluate the cell specific expression of fLuc mmRNA T-fLuc, leukocytes were isolated from the spleen 24 h post intravenous injection of tLNPs, sorted by the expression levels of Ly6c to Ly6c$^+$ and Ly6c$^-$ populations and fLuc expression was examined using luminometer (Fig. 2g, Supplementary Figure 2a–c, Supplementary Table 2). fLuc expression demonstrated ~100-fold increase in fLuc signal in Ly6c+ leukocytes using T-fLuc compared to I-fLuc treated mice and no significant difference was evaluated in the Ly6c- leukocytes population. These results are, to the best of our knowledge, the first report of targeted cell specific expression of mmRNA-LNPs in vivo.

Encouraged by the specificity of tLNPs, we have formulated mmRNA encoding IL-10 in tLNPs (T-IL10). T-IL10 or T-fLuc as a control were intravenously injected to DSS induced mice. IL10 protein expression was measured in the primary site of inflammation, the intestine and in the spleen 24 h post-injection by ELISA. Increased levels of IL10 were detected in both the intestine and spleen in T-IL10 treated mice compared to T-fLuc injected mice, demonstrating a specific IL10 expression in inflammation related organs. These results highlighted the possibility to express anti-inflammatory cytokines selectively in a specific cell type in vivo (Fig. 2h, Supplementary Table 2).

**Safety profile analysis of mmRNA tLNPs.** Safety is a key element prior to the translation of LNPs into the clinic in general and tLNPs with RNA payloads in particular. We have evaluated liver toxicity and immune-profiling of tLNPs. Liver toxicity was

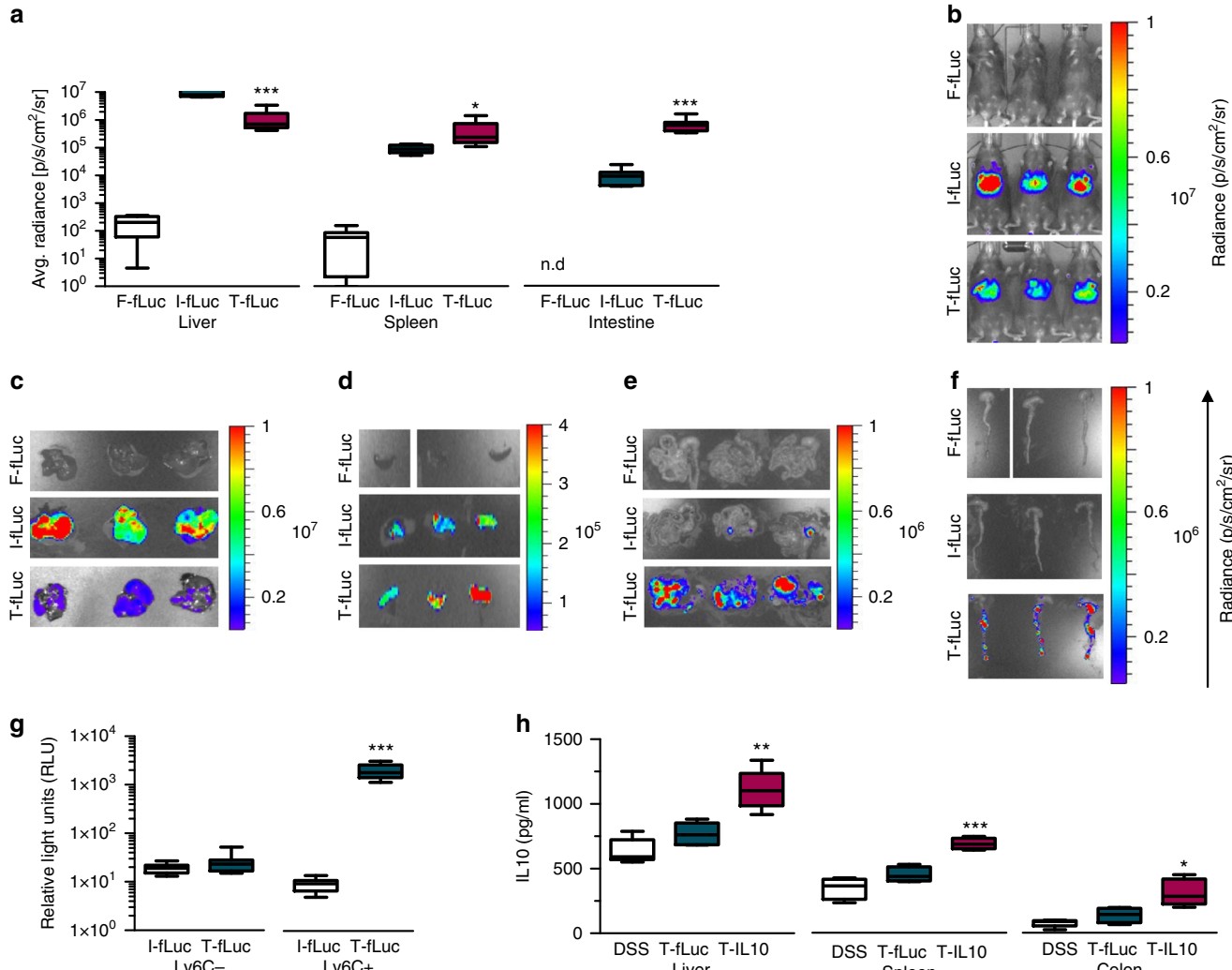

**Fig. 2** Specific protein expression in Ly6c cells in vivo. **a** Bioluminescence analysis of fLuc expression per organ as quantified using IVIS SpectrumCT and Living Image software. Significance display (**a**) were measured using two-sided Student's *t*-test between I-fLuc and T-fLuc, *n* = 8. **b-f** Representative IVIS SpectrumCT images from 3 mice that were injected with free fLuc mmRNA (F-fLuc), IgG I-fLuc LNPs or aLy6C T-fLuc LNPs, whole mice (**b**), livers (**c**), spleens (**d**) intestine (**e**), and colon (**f**). **g** Bioluminescence analysis of Ly6C+ and Ly6C- sorted leukocytes, isolated from mice's spleen, 24 h after injection of I-LNPs or aLy6C tLNPs; Significance display (**g**) were measured using two-sided Student's *t*-test between I-fLuc and T-fLuc, *n* = 10. **h** IL10 expression per organ, correlated with T-fLuc and T-IL10 treatments, as analyzed from a whole organ lysate using ELISA. Significance display (**h**) were measured using students t-test by comparing T-fLuc and T-IL10, *n* = 5. Scale bars are as follow, 0–1 × 10$^7$ radiance (**b, c**), 0.5–4 × 10$^5$ radiance (**d**), 0–1 × 10$^6$ radiance (**e, f**). Data presented as IQR with a median center line and min to max error bars, *$p$ < 0.05, **$p$ < 0.01, ***$p$ < 0.001, Data (**a–g**) are representative of three independent experiments

evaluated by liver enzymes elevation in the blood (alanine transaminase (ALT), aspartate aminotransferase (AST), and alkaline phosphatase (ALP)), and by liver histology. fLuc, T-fLuc, I-IL10, or T-IL10 LNPs were injected intravenously and the blood levels of ALT, AST, and ALP were measured 24 h post-injection concomitantly liver samples were sent to blinded histological analysis. Upon treatment, liver enzyme levels remained in the normal range in all treatment groups (Fig. 3a, Supplementary Table 3). Furthermore, liver histology did not reveal necrosis, massive bleeding or other morphological changes (Fig. 3b). These results demonstrate that treatment with mmRNA tLNPs do not cause liver toxicity nor affect liver's morphology. Immunogenicity was evaluated by measuring mice blood counts and by the measurements of splenic pro-inflammatory cytokines TNFα and IL6 by ELISA 24 h post intravenous tLNPs injection. No signs of immunogenicity were noticed in blood count (Fig. 3c, d, Supplementary Figure 3c, Supplementary Table 3) and in splenic IL6 and TNFα cytokine analysis of all treatment groups (Supplementary Figure 3a). Altogether, the mmRNAs-tLNPs strategy was found to be a safe, non-toxic and non-immunogenic in these experiments.

### tLNPs mediate therapeutic IL10 expression in colitis model.

Upon demonstration of specific IL-10 expression in inflammation related Ly6c[+] leukocytes in vivo, we explored the therapeutic potential of IL-10 expression in Ly6c[+] leukocytes in vivo in DSS induced colitis mice (Fig. 4a). Targeted expression of anti-inflammatory mediators, such as IL10, is expected to reduce inflammation specifically in the site of inflammation while minimizing systemic effect, thus improving the therapeutic effect with reduced toxicity. T-IL10, T-fLuc or I-IL10 LNPs were injected intravenously on days 3, 6, and 9 after colitis induction with DSS (Fig. 4b). Spleenic TNFα and IL6 cytokine levels were assessed in order to rule out LNP's immunogenicity (Supplementary Figure 3a–b). In order to assess inflammation severity, we analyzed cytokine levels in the colon by ELISA, as well as Colitis related pathological signs. Cytokine analysis in the colon

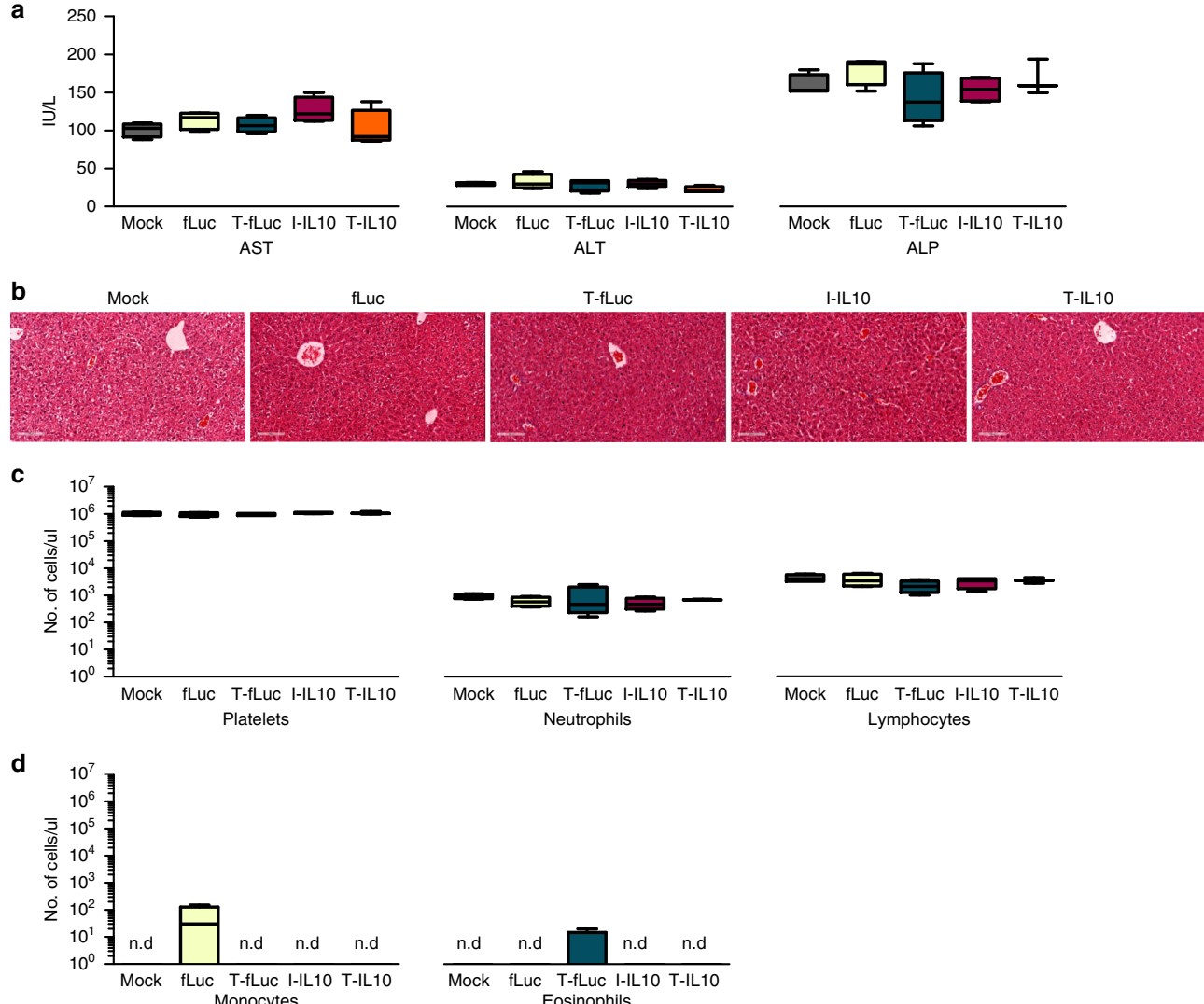

**Fig. 3** Safety profile analysis of mmRNA tLNPs. mmRNA LNPs safety profile as assessed 24 h post i.v administration using biochemistry analysis of blood samples (**a**); one-way ANOVA with Dunnett's multiple comparison, comparing all groups to mock treated mice; $n = 4$, ns. Histology of liver samples (hematoxylin and eosin staining, representative images) (**b**) and complete blood count (**c**, **d**) showing all positive indicators. Statistical analysis was performed using one-way ANOVA with Dunnett's multiple comparison, comparing all groups to mock treated mice; $n = 4$, ns. Data presented as IQR with a median center line and min to max error bars, ns denote $p > 0.05$, representative of three independent experiments (biological replicates)

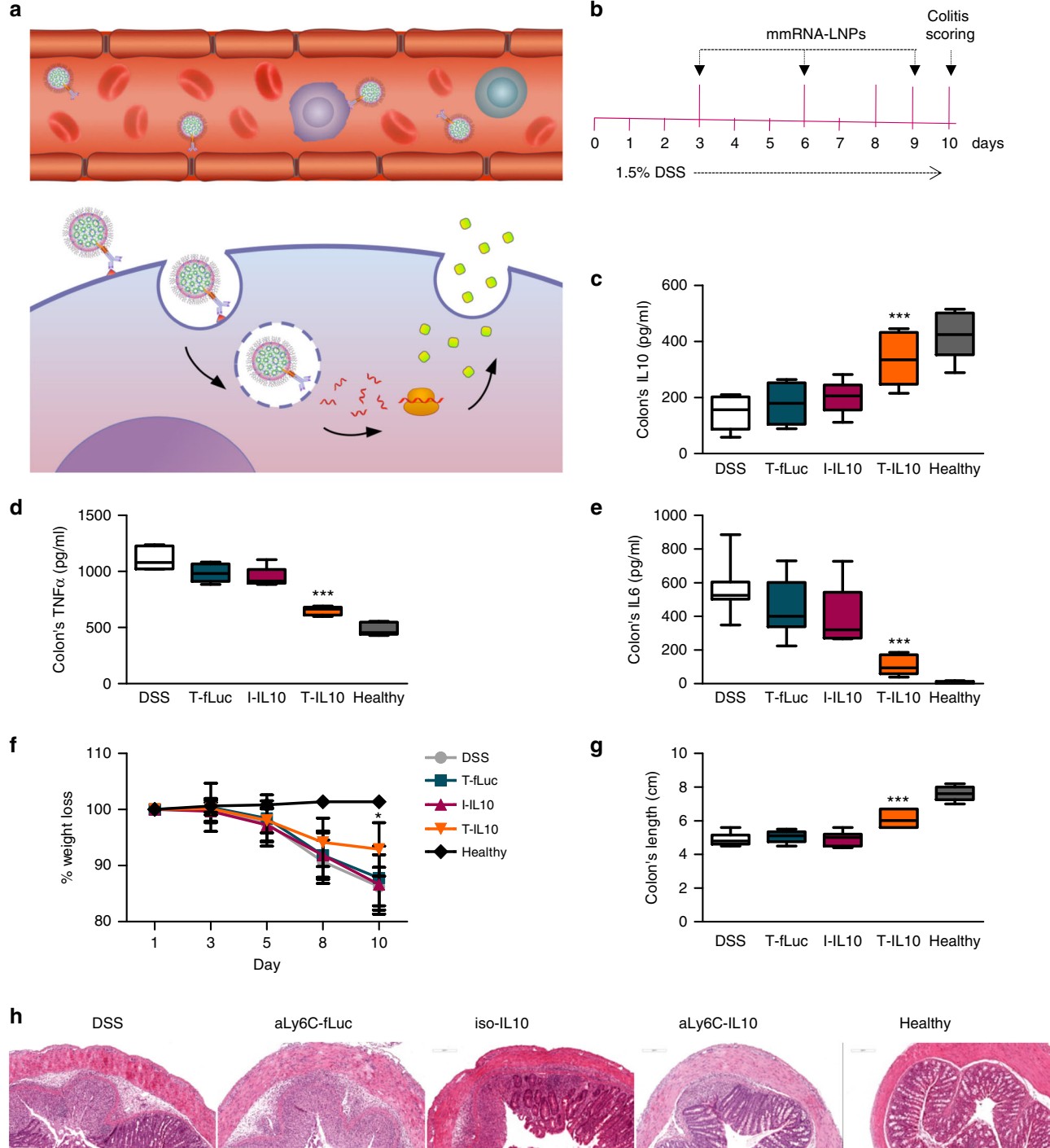

**Fig. 4** Selective therapeutic IL10 expression in DSS model. **a** Schematic illustration of tLNPs in the blood stream (1); tLNPs binding to the receptor (2); internalization (3) and mmRNA endosomal release (4); followed by an expression (5) and secretion of IL10 protein (6) **b** Experimental design. After beginning oral DSS, mice were injected intravenously with αLy6C or isotype control tLNP encapsulating IL10 or fLuc mmRNAs on days 3, 6, and 9 and were sacrificed on day 10. Colonic IL10 (**c**) TNFα (**d**) and IL-6 (**e**) levels in whole tissue lysates were evaluated by ELISA. Body weight was measured daily (**g**) and colon length (**f**) was assessed on day 10. Representative histology images (hematoxylin and eosin staining) of samples taken on day 10 (**h**). Data presented are IQR with a median center line and min to max error bars (**c–e**, **g**) and mean ± s.d. of two independent experiments as a representative of four independent experiments (biological replicates). $n = 9$, *$p < 0.05$, **$p < 0.01$, ***$p < 0.001$, one-way ANOVA with Dunnett's multiple comparison post hoc test to compare T-IL10 group with all other control groups. Statistical analysis in **f** was calculated using one-way ANOVA with Dunnett's multiple comparison post-hoc test to compare T-IL10 group with all other control groups on day 10

revealed significantly higher IL10 levels, as well as a lower amount of TNFα and IL6 pro-inflammatory cytokines in T-IL10 compared to T-fLuc and I-IL10 control groups (Fig. 4c–e, Supplementary Table 4). DSS colitis leads to severe pathological signs including weight loss, a shortening of the inflamed colon and erythema, swelling and inflammatory infiltration of the colon. All of these disease signs were significantly and dramatically reduced in T-IL10 treated mice as assessed by colonoscopy, colon's length

measurements and histology (Fig. 4g, h, Supplementary Table 4). In contrast, there was minimal reduction in T-fLuc and I-IL10 control groups. These results highlight the therapeutic potential of targeted expression using mmRNA-LNPs of anti-inflammatory cytokines in the inflammation related cells as a novel therapeutic strategy for inflammation related diseases.

## Discussion

Recombinant proteins are commonly used as therapeutic agents systemically and locally in various diseases, for instance as immuno-modulators, tumor suppressors and for the replacement of defected proteins[20]. Traditionally, when a recombinant protein serves as therapeutic agent, frequent injections are needed to maintain protein concentration in the therapeutic window[16,17]. Furthermore, high cost and long purification processes of recombinant protein production limit their use in the pharmaceutical industry. Taken together, the use of recombinant proteins as therapeutic agents is limited due to therapeutic and technological constraints, and thus a suitable safe, alternative is needed. mmRNA based protein expression is a novel strategy for prevention and treatment of various human diseases. However, the major limitation of this strategy is the lack of effective and specific delivery approaches. Furthermore, nonspecific and systemic protein expression can result in high toxicity, adverse effects and low therapeutic benefit. Therefore, targeted protein expression can provide a valuable tool for controllable and selective protein expression, with lower off target expression and reduced toxicity. Here we report on the first in vivo targeted delivery system for selective and efficient mmRNA based protein expression in leukocytes. We validated our targeting specificity using fLuc reporter gene and demonstrated a subset specific protein expression with an increased expression in the inflammation site and reduced expression in the liver. While a substantial decrease in off-target protein expression was achieved, an optimization is still required. We have demonstrated a specific IL10 expression in the diseased tissues by disease related Ly6c$^+$ inflammatory leukocytes. A selective expression of anti-inflammatory therapeutic proteins by inflammatory leukocytes, located mainly in the inflamed tissue, can enable an accurate tuning of the immune response, without a destructive systemic immune suppression. By targeted expression of the anti-inflammatory IL10, we were able to achieve therapeutic concentrations of this cytokine in the colon, while reducing off target protein expression. Compared to the rapid clearance of the recombinant protein, we were able to demonstrate a profound and extended IL10 expression lasted for more than 48 h. Intestinal IL10 expression resulted in a significant reduction in colitis related pathological symptoms and in the severity of intestinal inflammation.

## Methods

**Monoclonal antibodies**. αLy6c (clone Monts1, BE0203, BioXcell, USA) Isotype control (clone 2A3, BE0089, BioXcell, USA).

αLy6C (clone HK1.4, 128022, BioLegend, USA).

**Cell lines**. RAW 264.7 cells (ATCC, TIB-71) were routinely checked every two months for Mycoplasma contamination using EZ-PCR™ Mycoplasma Test Kit (Biological Industries, Israel) by manufacture protocol.

**mmRNAs**. Polyadenylated and capped mmRNAs were synthesized by TriLink (San Diego, CA) as followed:

CleanCap$^{TM}$ Firefly luciferase mmRNA

Custom CleanCap$^{TM}$ IL10 mmRNA (based on NM_010548.2):
atgcctggctcagcactgctatgctgcctgctctcttactgactggcatgaggatcagcaggggccagtacagccggaa gacaataactgcacccacttcccagtcggccagagccacatgctcctagagctgcggactgcctcagccaggtgaa gactttctttcaaacaaaggaccagctggacaacatactgctaaccgactccttaatgcaggactttaagggt tacttgggttgccaagccttatcggaaatgatccagttttacctggtagaagtgatgcccaggcagagaagcatggcc cagaaatcaaggagcatttgaattcctgggtgagaagctgaagaccctcaggatgcggctgaggcgctgtcatc

gatttctcccctgtgaaaataagagcaaggcagtggagcaggtgaagagtgatttttaataagctccaagaccaaggtgtc tacaaggccatgaatgaatttgacatcttcatcaactgcatagaagcatacatgatgatcaaaatgaaaagctaa

**Lipids**. DSPC, Cholesterol, DMG-PEG, and DSPE-PEG (Avanti Polar Lipids, USA).

Dlin-MC3-DMA (MC3) was synthesized according to a previously described method[21]. Shortly, Linoleic alcohol (10 g, 37 mmol) was dissolved in 50 mL DCM. Triethylamine (5.7 g, 56 mmol) was added and the solution was cooled in an ice-bath. Mesyl chloride (5.2 g, 49 mmol) in dichloromethane (50 mL) was added dropwise to the mixture. The reaction mixture allowed to warm to room temperature and stirred overnight. The reaction mixture was diluted with 200 mL DCM, washed with water followed by saturated NaHCO$_3$ and brine solution. The organic layer was dried over anhydrous NaSO$_4$ and the solvent was removed. The crude product was purified by column chromatography using silica gel for a pure compound (12 g, 90%). Mesylate (12 g, 34 mmol) was dissolved in anhydrous ether (50 mL). MgBr·Et2O (27 g, 104 mmol) was added to the reaction mixture under argon and the mixture was refluxed under argon for 24 h. The reaction mixture was diluted with ether (200 mL) and washed with ice-cold water (200 mL). The organic layer was further washed with brine solution (100 mL) and dried over anhydrous Na$_2$SO$_4$. The organic layer was then concentrated and the crude product was purified by column chromatography using silica gel to obtain a pure compound (11.5 g, 94%). Anhydrous ether (5 mL) was added to freshly activated Mg turnings (0.4 g, 18 mmol, argon), followed a dropwise addition of the bromide compound synthesized in above step (3 g, 9 mmol), dissolved in 5 mL of anhydrous ether, while cooling the flask in water. An exothermic reaction was observed as the indication for initiation of the reaction. The reaction mixture was kept at RT for 1 h and then cooled in ice bath. Ethyl formate (0.35 g, 4.5 mmol), dissolved in 5 mL of anhydrous ether, was added dropwise to the reaction mixture. An exothermic reaction was observed as the indication for initiation of the reaction. Then, the rest of the solution was quickly added as a stream and the reaction mixture was stirred for 1 h at room temperature. The reaction was quenched by adding ice cold water (20 mL). The reaction mixture was treated with aq. H2SO4 (10% by volume, 20 mL) until the solution became homogeneous and the layers were separated. The aqueous phase was extracted with ether (2 × 50 mL). The combined organic layers were dried over anhydrous Na$_2$SO$_4$ and concentrated. The crude product was purified by column chromatography using silica gel and the pure product was obtained as colorless liquid (1.2 g, 65%). The above compound (1.2 g, 2.3 mmol) was dissolved in 20 mL of dry DCM and dimethylaminobutyric acid (0.55 g, 3.4 mmol) was added followed by the addition of EDCI (1.3 g, 6.8 mmol) and cat amount of DMAP. The reaction mixture was stirred at room temperature overnight. The reaction mixture was washed with saturated NaHCO3 followed by water and brine solution. The organic layer was dried over anhydrous Na$_2$SO$_4$ and the solvents were removed. The crude product was purified by column chromatography using silica gel to a pure colorless liquid (1 g, 70%).

**Preparation of LNP entrapping mmRNAs**. LNPs were synthesized by mixing one volume of lipid mixture of MC3, DSPC, Cholesterol, DMG-PEG, and DSPE-PEG (50:10.5:38:1.4:0.1 mol ratio) in ethanol and three volumes of mmRNA (1:16 w/w mmRNA to lipid) in acetate buffer were injected in to a micro fluidic mixing device Nanoassemblr® (Precision Nanosystems, Vancouver BC) at a combined flow rate of 2 mL/min (0.5 mL/min for ethanol and 1.5 mL/min for aqueous buffer). The resultant mixture was dialyzed against phosphate buffered saline (PBS) (pH 7.4) for 16 h to remove ethanol.

**Size distribution**. LNP sizes in PBS were measured by dynamic light scattering using a Malvern nano-ZS Zetasizer (Malvern Instruments Ltd., Worcestershire, UK).

**ASSET expression and purification**. E. coli BL21-tuner (DE3) cells were transformed with the ASSET expression vector. When OD600 nm reached 1, 0.5 mM IPTG was added for induction O.N. at 30 °C. ASSET was purified from the membrane fraction and solubilized in 20 mM TRIS(HCL) (pH 8) buffer with 1% Triton™ X-100 (Sigma-Aldrich, Israel), followed by buffer exchange to 1.4% Octyl glucoside (Abcam, USA ab-142071-50-B)[22,23] and further purified using HisTrap HP columns (GE Healthcare Life Science). To create micelles, 250 nM cholesterol (Avanti Polar Lipids, USA) was added and ASSET was stored at −20 °C.

**Transmission electron microscopy**. A drop of aqueous solution containing LNP or ASSET LNP was placed on a carbon-coated copper grid and dried and analyzed using a JEOL 1200 EX (Japan) transmission electron microscope.

**ASSET LNP incorporation and tLNP assembly**. ASSET was incorporated into LNPs by an incubation of ASSET with LNPs for 48 h at 4 °C. To construct tLNP, RIg was incubated with ASSET LNP for 30 min (1:1, RIg:ASSET weight ratio).

**mmRNA entrapment efficiency**. mmRNA encapsulation efficiency was determined by agarose gel electrophoresis. Briefly, the encapsulation efficiency was

determined by comparing Ethidium bromide mmRNA staining of LNP and ASSET LNP in the presence or absence of 0.2% Triton™ X-100 (Sigma-Aldrich).

**Quantification of mmRNA encapsulation**. To quantify LNP after conjugation or ASSET assembly procedure, Quant-iT™ RiboGreen® RNA assay (Life Technology, CA, USA) was used. Two microliter of LNP or dilutions of mmRNA at known concentrations were diluted in a final volume of 100 μL of TE buffer (10 mM Tris-HCl, 20 mM EDTA) with or without 1% Triton™ X-100 (Sigma-Aldrich) in a 96-well fluorescent plate (Costar®, Corning®, NY, USA). The plate was incubated for 10 min at 40 °C to allow particles to become permeabilized before adding 99 μL of TE buffer and 1 μL of RiboGreen® reagent to each well. Plates were shaken at room temperature for 5 min and fluorescence (ex −485 nm, em −528 nm) was measured using a plate reader (Biotek).

**In vitro IL10 mmRNA expression**. RAW 264.7 cells (ATCC, TIB-71) (80% confluence) were incubated with 5 μg/mL of IL10 or fLuc mmRNA entrapped in LNPs or Lipofectamine MessengerMAX transfection reagent as control (Thermo Fisher). 24–48 h after LNPs' treatment IL10 concentration in the growth media was analyzed using ELISA (R&D Systems, USA).

**Animal experiments**. All animal protocols were approved by Tel Aviv University Institutional Animal Care and Usage Committee and in accordance with current regulations and standards of the Israel Ministry of Health. All animal experiments were conducted in a double blinded fashion; the researchers were blinded to group allocation and administered treatments. Mice were randomly divided in a blinded fashion in the beginning of each experiment.

**Ex vivo mmRNA expression**. To assess the feasibility of mmRNA loaded LNPs based protein expression in leukocytes we isolated Leukocytes from the spleen of 10-weeks-old female C57BL/6 mice (Harlan laboratories, Israel). In short, spleens were mashed on 70 μm cell strainer and further isolated from red blood cells using an osmotic shock, passed through a 70 μm cell strainer and washed twice with PBS. Isolated leukocytes, $2 \times 10^6$/mL, were growed in the presence of 1% non-essential amino acids, 1% sodium pyruvate and 0.1% β-mercaptoethanol (Rhenium, Israel), and incubated with 5 μg/mL IL10 or fLuc mmRNA encapsulated in LNPs for 24-48 h. IL10 expression was measured using ELISA kit (R&D Systems, USA) and fLuc expression was assessed using Luciferase Assay System (Promega, USA) and Veritas Microplate Luminometer (Turner BioSystems).

**In vivo bio-distribution**. Colitis was induced in 12-weeks-old female C57BL/6 mice (Harlan laboratories, Israel) using dextran sodium sulfate (DSS). Mice were fed for 10 days with 1.5% (wt/vol) DSS in the drinking water. To assess the selectivity of mmRNA expression, IL10 or fLuc mmRNA loaded LNPs were self-assembled with ASSET and αLy6C or Isotype control antibodies and were injected intravenously on day 5 from DSS treatment (1 mg/kg). On day 6, mice were injected intraperitoneal with 200 μL of 15 mg/mL XenoLight D-luciferin substrate (PerkinElmer, USA) and anesthetized using isoflurane (Piramal, Israel). Luminescence levels were assessed and analyzed for whole mice and different organs using IVIS SpectrumCT in vivo imaging system and Living Image software (PerkinElmer, USA). Colon and spleen samples were homogenized to assess cytokines by IL-6 and IL10 ELISA kits (R&D Systems, USA).

**In vivo mmRNA expression by LNPs**. Colitis was induced in 12-weeks-old female C57BL/6 mice (Harlan laboratories, Israel) using dextran sodium sulfate (DSS). Mice were fed for 10 days with 1.5% (wt/vol) DSS in the drinking water. To assess the selectivity of fLuc mmRNA expression, fLuc mmRNA loaded LNPs were self-assembled with ASSET and αLy6C or isotype control antibodies and were injected intravenously on day 5 from DSS treatment (1 mg/kg). On day 6, Leukocytes were isolated from the spleen, isolated from red blood cells using an osmotic shock, passed through a 70 μm cell strainer and washed twice with PBS. Cells in PBS containing 2% fetal bovine serum and 2 mM EDTA (Biological Industries, Israel) were stained with 1:100 fluorescently labeled αLy6C antibodies (BioLegend, USA) for 30 min at 4 °C and sorted for Ly6C+ and Ly6C− cells by flow cytometry (FACSAria, BD, USA). Sorted cells were lysed and fLuc expression was assessed in each leukocyte population using Luciferase Assay System (Promega, USA) and Veritas Microplate Luminometer (Turner BioSystems).

**Efficacy in IBD model**. Colitis was induced in 12-weeks-old female C57BL/6 mice (Harlan laboratories, Israel) using dextran sodium sulfate (DSS). Mice were fed for 10 days with 1.5% (wt/vol) DSS in the drinking water. Suspensions (200 μL in PBS) of LNP loaded with 1 mg/kg mmRNAs and self-assembled with αLy6C or isotype control RIg (BioXcell, USA) were injected intravenously on days 3, 6, and 9. Body weight was monitored every other day. On day 10 mice were sacrificed and colitis severity was assessed. The length of the entire colon from cecum to anus was measured. Small segments of the colon were taken for histologic and immuno-histochemistry evaluation. Colon, liver and spleen samples were homogenized to assess cytokines by IL-6 and IL10 ELISA kits (R&D Systems, USA).

**In vivo toxicity**. Ten-weeks-old female C57BL/6 mice (Harlan laboratories) were injected with 1 mg/kg fLuc mmRNA or IL10 mmRNA encapsulated in LNPs that were incorporated with ASSET and isotype control or αLy6C antibodies. Twenty-four hours after injection blood was collected for biochemistry using Cobas-6000 instrument and complete blood count via Sysmex and Advia-120 (A.M.L, Israel) and liver samples were taken to histology (Histospeck, Israel).

**Statistical analysis**. All data are expressed as median ± min to max or mean ± s.d. Statistical analysis for comparing two experimental groups was performed using two-sided Student's $t$-test. In experiments with multiple groups we used one-way ANOVA with Dunnett's multiple comparison post-hoc test. A value of $p < 0.05$ was considered statistically significant. Analyses were performed with Prism 5 (Graph pad Software). Differences are labeled $*p \leq 0.05$, $**p \leq 0.01$, $***p \leq 0.001$, and $****p \leq 0.0001$. Sample size of each experiment was determined to be the minimal necessary for statistical significance by the common practice in the field. Similarity between variances of each statistically compared groups were verified by $F$-test. Pre-established criteria for removal of animals from experiment were based on animal health, behavior and well-being as required by ethical guidelines; no animals were excluded from the experiments.

## Data availability
All relevant data are available from the authors upon reasonable request.

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

## Acknowledgements

N.V. thanks the Marian Gertner Institute of Medical Nanosystems, the Dan David Fellowship Award and Iafa Keidar Prize for all their support. This work was supported in part by grants from The Dotan Center for Hemato-Oncology at Tel Aviv University; The Lewis Trust for Blood Cancer; and by the ERC grant LeukoTheranostics (# 647410) awarded to DP.

## Author contributions

N.V. and D.P. conceived the study. N.V., M.G., Y.G., N.D., D.R. R.K., S.R. performed the research and analyzed the data. N.V., D.R., and D.P. wrote the manuscript

## Additional information

**Competing interests:** D.P. declare financial interest in Quiet Therapeutics, ART Biosciences and SEPL pharma. The remaining authors declare no competing interests.

