## [Peer Review File · Nature Communications]

Reviewers' comments:

Reviewer #1 (Remarks to the Author):

The authors have developed LNPs using ASSET system and Anti-Ly6c antibody coating to promote selective mRNA delivery in leukocytes in vivo. They have shown the efficacy of the formulations using a reporter fLuc mmRNA and therapeutic IL-10 mmRNA in DSS colitis model. The authors have demonstrated a profile analysis of the LNPs in vivo. The study design is systematic and nicely articulated.

However, there are several concerns with experimental controls and analysis as mentioned below:

1. In Figure 1, schematic representations, because of the absence of labels for different moieties, it is difficult to understand overview of the outline. In the characterization of LNPs, what is the charge of the LNPs formed with mmRNA and how does it help in cellular delivery?
2. Figure 1, for in vitro experiments, I did not find any detail for number of cells used for the mentioned dose, also it is better to include dose in molarity than microg/ml for clarity. How is the luc or IL-10 expression normalized when comparing different groups as units mentioned are pg/ml or RLU. There is no positive control in cell experiments (Fig. 1, d and e), cannot understand the efficacy of this system. Figure legends should also include end time points and doses used for in vitro or ex vivo experiments.
3. Figure 2 b, for IVIS imaging, the control group is absent to compare to the efficacy of LNPs loaded with either T-fluc or I-fluc. It would be good to include some control group, whether its naked mmRNA or even untreated group. Also, the whole body IVIS is showing high luminescence for I-fluc group at least to the naked eye? However, in Figure 2a, T-fluc shows higher bioluminescence for spleen and intestine. Is something wrong in either of the figures?
4. Figure 2a, is there a difference in luc-mmRNA expression in different regions of the intestine (distal or proximal)? In Figure 2 g, the significance in colon T-IL10 group is it compared to DSS group or T-fluc. It is not clear in legend. Also, from looking at overlapping error bars between these groups, it is hard to imagine a difference between T-fluc and T-IL10 group in colon, although authors mention in text "Increased levels of IL10 were detected in both the intestine and spleen in T-IL10 212 treated mice compared to T-fLuc injected mice, demonstrating a specific IL10 213 expression in inflammation-related organs". Could you clarify this point? In this regard, how are reporter group and IL-10 results co-related in terms of organ distribution?
5. Figure 4, Authors mention "Colon's cytokine analysis revealed significantly higher IL10 levels as well as 269 lower amount of TNF α and IL6 pro-inflammatory cytokines in T-IL10 compared to T-270 fLuc and I-IL10 control groups (Figure 4c-e)". However, in Figure 4c, I-IL10 and T-IL10 groups error bars seem to be overlapping, does not seem a significant difference. The detailed analysis should be shown in between the groups to show the difference between these two formulations.
6. Overall, in all the figures, IL-10 expression or bioluminescence should be normalized, pg/ml or just mentioning RLU is misleading, especially when authors are comparing different groups or organs for expression.
7. In statistical analysis, authors should clarify which groups are being compared either in legends or methods. * means between which two groups.

Reviewer #2 (Remarks to the Author):

This study by Veiga et al describes novel anti-Ly6c antibody coupled lipid nanoparticles (LNPs) containing modified mRNAs for IL-10. This group has extensive expertise in modifying nanoparticles with external antibodies and with delivering cargos to modulate immune functions. In this study, they demonstrate that systemically delivered anti-Ly6c antibody-coupled LNPs are able to enhance IL-10 expression in the livers, spleens, and colons of mice. They demonstrate that

liver function tests and histological markers are not perturbed in livers of recipient mice. Most importantly, they can significantly ameliorate mild DSS induced acute colitis.

The study is a technical advance and has moderate implications for gene therapy of experimental colitis. The authors find that IL-10 protein levels are elevated in livers, spleens, and colons of treated mice (Fig. 2g). Hence, the claim of intestine specific delivery of IL-10 is not well justified. Systemic delivery of these nanoparticles could in theory be therapeutically beneficial, but they may carry the same systemic side effects of other systemic immunosuppressive therapies (e.g., anti-TNF).

Letter to the reviewers

We would like to thank the reviewers for their in deep review and thoughtful comments. We have revised the manuscript according to the reviewers' comments and suggestions.

Reviewer #1 (Remarks to the Author):

The authors have developed LNPs using ASSET system and Anti-Ly6c antibody coating to promote selective mRNA delivery in leukocytes in vivo. They have shown the efficacy of the formulations using a reporter fLuc mmRNA and therapeutic IL-10 mmRNA in DSS colitis model. The authors have demonstrated a profile analysis of the LNPs in vivo. The study design is systematic and nicely articulated.

We thank the reviewer for the deep review and helpful comments.

However, there are several concerns with experimental controls and analysis as mentioned below:

1. In Figure 1, schematic representations, because of the absence of labels for different moieties, it is difficult to understand overview of the outline. In the characterization of LNPs, what is the charge of the LNPs formed with mmRNA and how does it help in cellular delivery?

Thank you very much for this important comment, we included a detailed description of the LNPs preparation in the revised version. Briefly, the LNPs are composed of a helper lipid, DSPC (1), PEG-lipid (2), ionizable amino lipid DLin-MC3-DMA (3) and cholesterol (4), diluted in ethanol and mmRNA (5) diluted in acidic acetate buffer (pH 4). Upon mixing in acidic condition, electrostatic interactions drive the formation of inverted micelles containing mmRNA molecules surrounded predominantly by the ionizable cationic lipid, DLin-MC3-DMA, which are then coated with PEG-lipids to form the LNPs. At the end, the pH is raised to physiological pH, leading to LNPs with neutral charge, which are more biocompatible (Supplementary Figure 1a). Subsequently, upon endocytosis to the acidic endosome, the amino group of the DLin-MC3-DMA is positively charged, which allows the release of mmRNA molecules to the cytosol.

2. Figure 1, for in vitro experiments, I did not find any detail for number of cells used for the mentioned dose, also it is better to include dose in molarity than microg/ml for clarity. How is the luc or IL-10 expression normalized when comparing different groups as units mentioned are pg/ml or RLU. There is no positive control in cell experiments (Fig. 1, d and e), cannot understand the efficacy of this system. Figure legends should also include end time points and doses used for in vitro or ex vivo experiments.

Thank you for these important comments. The legends of Figure 1 were revised and now include the number of cells. Protein dose, as analyzed by ELISA, is commonly displayed as nanogram/ml. Furthermore, we prefer the use of fLuc as a reporter mmRNA rather than to estimate the amount of protein that is been expressed. Unfortunately, leukocytes are among the most difficult to transfect cells, thus there is no

useful transfection reagent for primary leukocytes and by that we don't have a valid positive control. A positive control for an in-vitro transfection was added (Supplementary Figure 1e). In the revised manuscript we added the missing data to the figure legends.

3. Figure 2 b, for IVIS imaging, the control group is absent to compare to the efficacy of LNPs loaded with either T-fluc or I-fluc. It would be good to include some control group, whether its naked mmRNA or even untreated group. Also, the whole body IVIS is showing high luminescence for I-fluc group at least to the naked eye? However, in Figure2a, T-fluc shows higher bioluminescence for spleen and intestine. Is something wrong in either of the figures?

We thank the reviewer for raising this point. We repeated our biodistribution experiments with the addition of free mmRNA control mice. During all of the previous experiments we used untreated mice as a control.

The whole body IVIS signal might be misleading and might give a false image due to intrinsic limitations of the IVIS instrument. The whole mouse image is limited due to the location of each organ. It might be that the liver signal overpower the other close by organs, but it could also be a signal from the spleen and some of the intestine. Due to these limitations of the IVIS instrument we decided to analyze the organs after taking them out and measuring luminescence based on organs.

Unfortunately, there is no available methodology for analyzing depth and 3D localization in our facilities. Although the whole animal IVIS images are displayed, the ROIs were measured based on a single organ imaging.

4. Figure 2a, is there a difference in luc-mmRNA expression in different regions of the intestine (distal or proximal)? In Figure 2 g, the significance in colon T-IL10 group is it compared to DSS group or T-fluc. It is not clear in legend. Also, from looking at overlapping error bars between these groups, it is hard to imagine a difference between T-fluc and T-IL10 group in colon, although authors mention in text “ Increased levels of IL10 were detected in both the intestine and spleen in T-IL10 212 treated mice compared to T-fLuc injected mice, demonstrating a specific IL10 213 expression in inflammation-related organs”. Could you clarify this point? In this regard, how are reporter group and IL-10 results co-related in terms of organ distribution?

We thank the reviewer for these important remarks. fLuc reporter gene is expressed in a segmental characteristic, as can be noted by figure 2e-f, which we believe to correlates with the segmented nature of the intestinal inflammation in DSS colitis model. We didn't observe notable differences between distal and proximal regions of the intestine. Following your important comments we repeated our biodistribution experiments and emphasized the location of fLuc expression in the intestine (Figure 2e-f). The expression of fLuc reporter protein and IL10 protein correlates, although in the displayed graphs (Figure 2a, g) IL10 levels were analyzed only in the colon whereas fLuc expression was evaluated in the whole intestine. Unfortunately, due to technical limitations we can't analyze IL10 levels in the whole intestine. The statistical analysis was done by two-sided Student's *t*-test using Prism 5 Graph-pad Software, comparing between I-fLuc and T-

fLuc (figure 2a), and between T-fLuc and T-IL10 (figure 2h). These additions are now available in the revised manuscript.

5. Figure 4, Authors mention “Colon's cytokine analysis revealed significantly higher IL10 levels as well as 269 lower amount of TNF α and IL6 pro-inflammatory cytokines in T-IL10 compared to T- 270 fLuc and I-IL10 control groups (Figure 4c-e) “. However, in Figure 4c, I-IL10 and T-IL10 groups error bars seem to be overlapping, does not seem a significant difference. The detailed analysis should be shown in between the groups to show the difference between these two formulations.

Thank you for pointing this out. Statistical analysis was preformed using Prism 5 Graphpad Software. In experiments with multiple groups, as in Figure 4, we used one-way ANOVA with Dunnett's multiple comparison post hoc test to compare T-IL10 with all other control groups. Following your important comment, we added a sentence which clarify the statistical calculation.

6. Overall, in all the figures, IL-10 expression or bioluminescence should be normalized, pg/ml or just mentioning RLU is misleading, especially when authors are comparing different groups or organs for expression.

Thank you for the important remark. We prefer the use of fLuc as a reporter mmRNA rather than to estimate the amount of protein that is been expressed, indeed bioluminescence signal is caused by an enzymatic reaction and thus reinforced. We included a sentence regarding the nature of fLuc as an enzymatic reporter to clarify this point in the revised manuscript.

7. In statistical analysis, authors should clarify which groups are being compared either in legends or methods. * means between which two groups.

Thank you very much for the important note, we clarify the compared groups and wrote in in deep statistical analysis for each figure in a supplementary Table.

Reviewer #2 (Remarks to the Author):

This study by Veiga et al describes novel anti-Ly6c antibody coupled lipid nanoparticles (LNPs) containing modified mRNAs for IL-10. This group has extensive expertise in modifying nanoparticles with external antibodies and with delivering cargos to modulate immune functions. In this study, they demonstrate that systemically delivered anti-Ly6c antibody-coupled LNPs are able to enhance IL-10 expression in the livers, spleens, and colons of mice. They demonstrate that liver function tests and histological markers are not perturbed in livers of recipient mice. Most importantly, they can significantly ameliorate mild DSS induced acute colitis.

The study is a technical advance and has moderate implications for gene therapy of experimental colitis. The authors find that IL-10 protein levels are elevated in livers, spleens, and colons of treated mice (Fig. 2g). Hence, the claim of intestine specific delivery of IL-10 is not well justified. Systemic delivery of these nanoparticles could in theory be therapeutically beneficial, but they may carry the same systemic side effects of other systemic immunosuppressive therapies (e.g., anti-TNF).

We thank the reviewer for these enlightening remarks. Indeed, our targeting system is subset specific and not organ specific, with an increased expression in inflammation site and reduced liver expression. A selective expression of therapeutic proteins by inflammatory leukocytes, located mainly in the inflamed tissue, can enable an accurate tuning of the immune response. Additionally, we observed a significantly lower off target protein expression in the liver. Another important advantage is the half-life of IL10 protein. Compared to the rapid clearance of the recombinant protein, we were able to demonstrate a profound and extended IL10 expression lasted for more than 48 hours.

REVIEWERS' COMMENTS:

Reviewer #1 (Remarks to the Author):

The authors have addressed all of the prior criticisms. The manuscript can be accepted for publication.

Reviewer #2 (Remarks to the Author):

The revised manuscript is an important contribution to the literature.

--Averil Ma, MD